physical chemistry/thermodynamics/
environmental chemistry

fluorine-containing slag, volatilization, thermal analysis, steelmaking

**Author for correspondence:**
Junxue Zhao
e-mail: zhaojunxue1962@126.com

This article has been edited by the Royal Society of Chemistry, including the commissioning, peer review process and editorial aspects up to the point of acceptance.

# The volatilization behaviour of typical fluorine-containing slag in steelmaking

Zhongyu Zhao, Junxue Zhao, Zexin Tan, Boqiao Qu and Yaru Cui

School of Metallurgical Engineering, Xi'an University of Architecture and Technology, 13 Yanta Road, Xi'an, Shaanxi 710055. People's Republic of China

JZ, 0000-0002-5177-3244

It was taken as typical steelmaking fluorine-containing slag systems with the remelting electroslag, continuous casting mould flux and refining slag. The volatilization behaviour of each slag system was analysed by thermogravimetric (TG) and mass spectrometry (MS) detection. The results showed that the remelting electroslag volatilized significantly above 1300°C and the volatiles were mainly $CaF_2$, $MgF_2$ with a small amount of $SiF_4$ and $AlF_3$; the continuous casting mould flux volatilization was divided into two stages, in the first stage (500°C~800°C), $CaF_2$ and $Na_2O$ reacted to form NaF, and in the second stage (greater than 1200°C), the $CaF_2$ evaporation was highlighted; for $CaF_2$-CaO-based refining slag, the volatilization was the most significant at the eutectic point 84% $CaF_2$–16% CaO, and the volatility can be reduced by adding 5% $SiO_2$. This research will be guiding significance for the composition and performance control of fluorine-containing slag and metallurgical environmental protection in the steelmaking process.

## 1. Introduction

The steelmaking process is actually a 'slag-making' process. The physico-chemical properties of slag are of significant effects on the melting temperature, chemical reactions, metal solidification and inclusions removal. Therefore, the slag composition must be reasonably controlled to meet different melting requirements [1]. The fluoride and potassium sodium oxides are widely used as fluxes to satisfy the slag high-temperature physico-chemical properties, and the form and amount of fluoride depend on the production indexes of different steel plants and the performance requirements of different steel. It is generally known that the steelmaking process is mostly in heating or holding process to maintain the slag–metal reaction. The slag will volatilize at high temperature and eventually change the slag composition and metallurgical properties, if the slag contains fluoride or alkali metal oxide ($Na_2O$, $K_2O$) [2–12].

**Figure 1.** Fluorine-containing slag. (*a*) Remelting electroslag, E1; (*b*) remelting electroslag, E2, E3; (*c*) remelting electroslag, E4; (*d*) continuous casting mould flux.

On these issues, a large number of scholars have done some research on the slag volatilization. Mills [13] and Mao [2] judged that the volatiles were NaF, KF, $SiF_4$, $AlF_3$ and $CaF_2$ by thermodynamic calculation. Chen and Liang [14–17] simulated the fluoride volatilization process by establishing kinetic models. Zhao *et al.* [18] analysed the important influence of fluoride on the properties of mould flux, such as melting point, viscosity and crystalline. Shang *et al.* [19] summarized the influence of slag volatilization characteristics on the physico-chemical properties by measuring the melting point and viscosity of fluorine-containing electroslag. However, the study around volatiles is still limited to the theoretical calculation or indirect experimental analysis, and the conclusions are not convincing. The qualitative and quantitative analysis is insufficient. In this paper, three kinds of typical fluorine-containing slag systems for steelmaking process will be investigated and compared with the volatilization characteristics by thermogravimetric (TG), mass spectrometry (MS) and X-ray fluorescence (XRF) methods, which can directly reflect the volatilization characteristics of different steelmaking slag. It can be practical and application value of slag composition and metallurgical performance control in steelmaking.

## 2. Materials and methods

In order to systematically analyse the slag volatilization behaviour in the steelmaking process, three typical steelmaking fluorine-containing slag systems were selected from Taiyuan Iron&Steel (Group) Co., Ltd, of which the annual production capacity is 0.3 million tons of steel ingots, including the remelting electroslag, the continuous casting mould flux and the traditional refining slag based on $CaF_2$-CaO, shown in figure 1.

The volatilization characteristics of each slag system were analysed by TG-MS. The specific research methods were described as follows:

— Take XRF-1800 by melt press in Pt-Rh crucible to detect the slag composition and prepare different slag samples with chemical reagents according to XRF results. The samples were ground by an agate ball mill at a speed of 200 r.p.m. for 0.5 h, dried at 373 K for 5 h and then sealed and stored in the dark.
— For the remelting electroslag and continuous casting mould flux, take TG-MS tests to determine the volatilization temperature, different volatiles and volatilization ratio (take NETZSCH 449-F3 analyser for TG test and STA 409 C/CD mass spectrometer for MS test); for $CaF_2$-CaO-based refining slag, it was prepared with chemical pure reagents according to different proportions. The reagent

**Table 1.** The reagent information.

| reagent | purity % | granularity | batch |
|---|---|---|---|
| $CaF_2$ | ≥98.5 | white powder | 20160109 |
| $CaO$ | ≥98.0 | white powder | 20160807 |
| $SiO_2$ | ≥99.0 | (0.65 ∼ 0.85 mm) ≥ 89.0% | 20160819 |
| $Al_2O_3$ (neutral) | >99.0 | white powder | 20160814 |
| $MgO$ (light) | ≥98.5 | white powder | 20160709 |
| $Na_2CO_3$ | ≥99.8 | white powder | 20161022 |

**Table 2.** Components of remelting electroslag and the weight loss in TG test (%).

| no. | $CaF_2$ | $CaO$ | $SiO_2$ | $Al_2O_3$ | $MgO$ | weight loss at 1450°C |
|---|---|---|---|---|---|---|
| E1 | 39.64 | 24.82 | 10.25 | 18.31 | 6.98 | 7.1 |
| E2 | 31.52 | 32.05 | 11.65 | 17.66 | 7.12 | 4.8 |
| E3 | 32.06 | 24.59 | 19.15 | 17.15 | 7.05 | 5.5 |
| E4 | 27.35 | 35.06 | 9.64 | 17.93 | 10.02 | 1.8 |

information is shown in table 1. The weight loss of each sample was recorded by constant temperature heating in a tubular furnace. The experimental parameters were set as follows:

(i) TG test: the heating rate was 10°C min$^{-1}$ with Ar gas at a flow rate of 50 ml min$^{-1}$;

(ii) MS test: the heating rate was 10°C min$^{-1}$ (stage I); the temperature was maintained for 1 h at 1300°C (stage II) and then increased to 1400°C (stage III). Finally, the sample was slowly cooled (stage IV). It was protected with Ar gas at a flow rate of 50 ml min$^{-1}$.

(iii) Tubular furnace heating process: the refining slag samples mixed with reagents were prepared according to different proportions ($CaF_2$ pure reagent, 90% $CaF_2$–10% $CaO$, 84% $CaF_2$–16% $CaO$, 60% $CaF_2$–40% $CaO$, 40% $CaF_2$–60% $CaO$); tube furnace was set at 1500°C and samples were weighed within 5–30 min with Ar gas protection at a flow rate of 50 ml min$^{-1}$.

(Note: the platinum crucible was used in the TG-MS test and the graphite crucible was used in the tube furnace heating process.)

— The components of roasted samples were analysed by XRF for testing the above discussion on the volatilization characteristics of each slag system.

# 3. Results and discussion

## 3.1. Volatilization behaviour of remelting electroslag

Four kinds of remelting electroslag were analysed by XRF, as shown in table 2. The TG tests were carried out respectively, as shown in figure 2.

According to the weight loss curves in figure 2, the electroslag with different fluorine content had different weight loss, and the more the fluorine, the greater the weight loss rates. When the temperature was higher than 1300°C, the weight loss was obvious. It can be seen that the volatilization characteristics of remelting electroslag were directly related to $CaF_2$ and temperature.

To further explore the volatiles, the E3 sample was tested by MS and the results are shown in figure 3.

It can be seen from figure 3 that during the initial volatilization stage I (750°C–1200°C), the volatiles were divided into a small amount of $MgF_2$, $SiF_4$ and $AlF_3$, and from beginning to end, especially in high-temperature regions II–III (1300°C–1400°C), $CaF_2$ was the main volatile. This was basically consistent with the previous thermodynamic calculation results [14].

## 3.2. Volatilization behaviour of continuous casting mould flux

The continuous casting mould flux was mixed by chemical pure reagents and its composition is shown in table 3. The TG test is shown in figure 4.

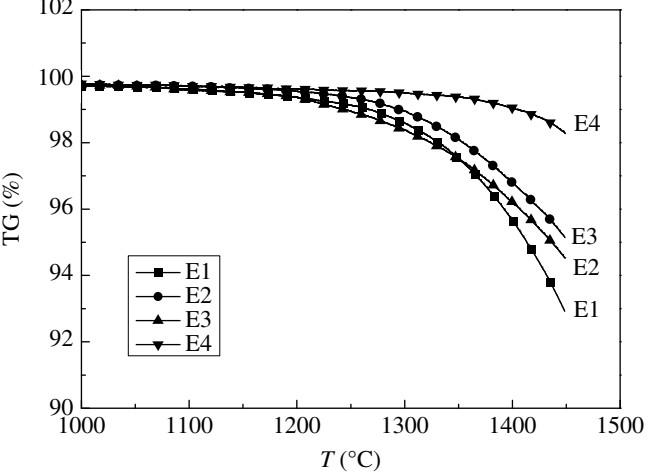

**Figure 2.** TG curves of remelting electroslag system.

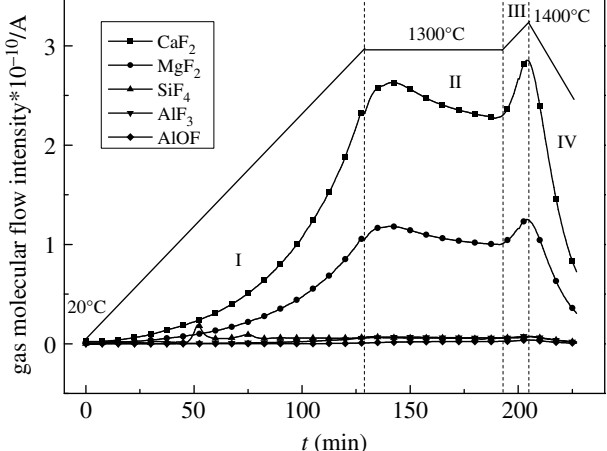

**Figure 3.** MS curves of remelting electroslag system (E3).

**Table 3.** Composition of the continuous casting mould flux (%).

| composition | $CaF_2$ | $Al_2O_3$ | MgO | $SiO_2$ | CaO | $Na_2CO_3$ | weight loss at 1400°C |
|---|---|---|---|---|---|---|---|
| quality | 17.6 | 3.8 | 3.0 | 27.2 | 27.2 | 21.2 | 16.0 |

According to TG curve in figure 4, the weight loss process could be divided into two stages: the first stage was from 500°C to 900°C and the weight loss rate was 9.7%; the second stage was above 1000°C, and the weight loss rate was 6.3%. According to the previous thermodynamic calculation under the same experimental conditions [18], the first stage was the reaction of $CaF_2$ with $Na_2O$ and $SiO_2$ to generate NaF and $SiF_4$ gas, and the second stage was mainly the $CaF_2$ evaporation, as shown in table 4. To determine the volatiles of the above samples, the MS test was carried out and is shown in figure 5.

From the MS curves in figure 5, it can be seen that the volatilization of continuous casting mould flux system was more complex than that of remelting electroslag. Although the gases such as $SiF_4$ and $MgF_2$ were generated at stage I from 500°C to 800°C, the NaF was volatilized preferentially and largely at stage II due to the strong activity of light metal oxide $Na_2O$. Similarly, when it was above 1200°C at stage III, the $CaF_2$ evaporation was highlighted.

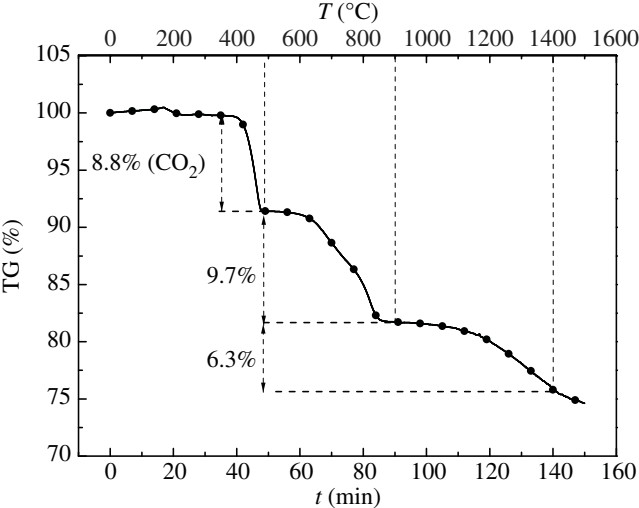

**Figure 4.** TG curve of the continuous casting mould flux. (Note: the weight loss before 500℃ could be ignored considering the decomposition reaction of $Na_2CO_3$, as follows. $Na_2CO_3 \triangleq Na_2O + CO_2\uparrow$).

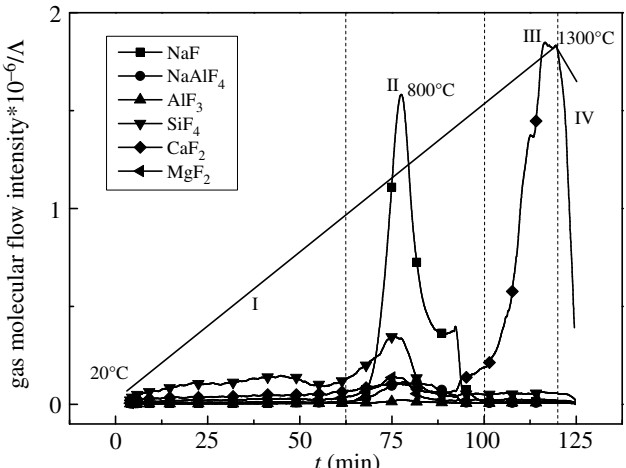

**Figure 5.** MS curves of the continuous casting mould flux.

**Table 4.** Volatiles reaction.

| no. | reaction |
|-----|----------|
| (1) | $CaF_2(s) = CaF_2(g)\uparrow$ |
| (2) | $CaF_2(s) + 1/2SiO_2(s) = 1/2SiF_4(g)\uparrow + CaO(s)$ |
| (3) | $CaF_2(s) + MgO(s) = MgF_2(g)\uparrow + CaO(s)$ |
| (4) | $CaF_2(s) + 1/3Al_2O_3(s) = CaO(s) + 10AlF_3(g)\uparrow$ |
| (5) | $CaF_2(s) + Na_2O(s) = 2NaF\ (g)\uparrow + CaO\ (g)$ |

## 3.3. Volatilization behaviour of $CaF_2$-CaO-based refining slag

The refining slag played a significant role in the steelmaking process for desulfurization and alloying. Different kinds of steel have different requirements for the properties and composition of refining slag. The traditional refining slag system is based on $CaF_2$-CaO, and sometimes added with an appropriate amount of $SiO_2$ (4%–11%) and $Al_2O_3$ (6%–9%). Therefore, focus on the volatilization behaviour, the $CaF_2$-CaO slag system was prepared with chemical reagents by different proportion and it could reveal the volatilization characteristics of traditional refining slag system.

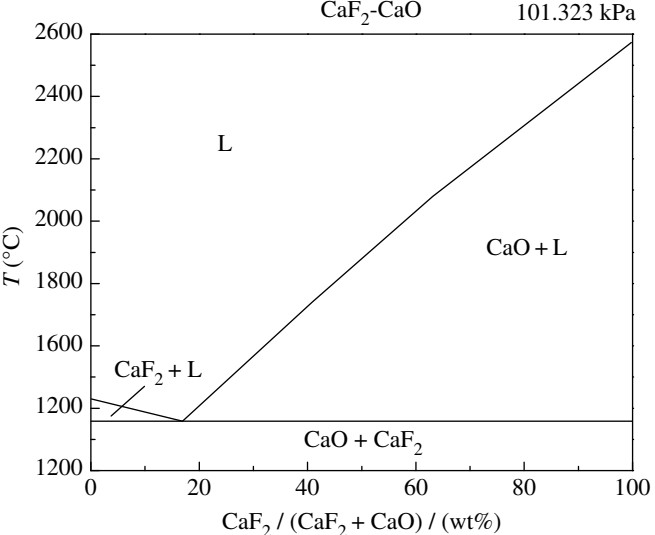

**Figure 6.** CaF₂-CaO phase diagram.

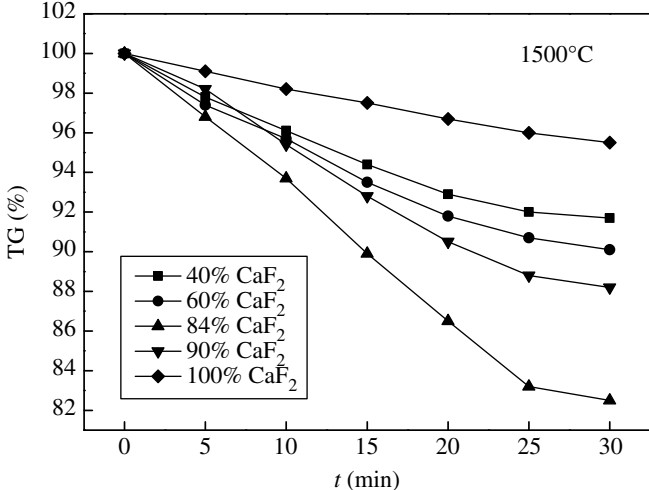

**Figure 7.** Effect of CaF₂ on volatilization of refining slag.

Firstly, the binary phase diagram of $CaF_2$-CaO was calculated by Factsage, as shown in figure 6, the eutectic point was 84% $CaF_2$–16% CaO. Then, the $CaF_2$-CaO-based refining slag with different composition was prepared by chemical reagents and heated in a tubular furnace at 1500°C. The weight loss of each sample after different holding time is shown in figure 7.

From figure 7 TG curves, it can be obtained that the volatilization of $CaF_2$-CaO slag was of the most significant at the eutectic point 84% $CaF_2$–16% CaO, and the volatile was $CaF_2$. Therefore, for this refining slag system, the $CaF_2$ proportion can be adjusted properly to weaken the slag volatilization. Furthermore, a small amount of $SiO_2$ (5%∼10%) was added to the $CaF_2$-CaO slag system to explore the effect of $SiO_2$ on volatilization at 1500°C, as shown in figure 8.

It can be seen from figure 8 that the volatility of the refining slag system can be reduced by adding a small amount of $SiO_2$ comparing R1 and R2. However, the $SiF_4$ gas would be generated to accelerate the volatility of the slag system if the $SiO_2$ was too much comparing R2, R3 and R4.

## 3.4. Examination of volatilization characteristics

As mentioned above, for the $CaF_2$-CaO-based refining slag, the volatile was single $CaF_2$, and no further inspection was required. For remelting electroslag and continuous casting mould flux, it was necessary to do XRF analysis of the samples after TG tests due to the complex volatilization characteristics, and this

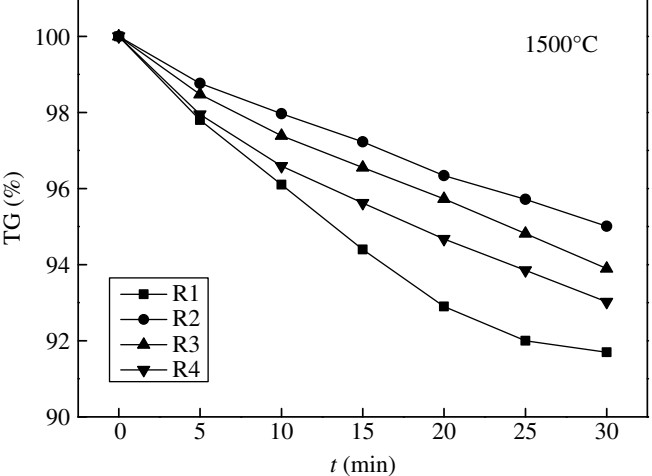

**Figure 8.** Effect of $SiO_2$ on volatilization of refining slag. (R1: 40% $CaF_2$–60% CaO; R2: 38% $CaF_2$–57% CaO–5% $SiO_2$; R3: 36% $CaF_2$–54% CaO–10% $SiO_2$; R4: 33% $CaF_2$–50% CaO–17% $SiO_2$).

**Table 5.** Components of remelting electroslag after TG tests (%).

| no. | $CaF_2$ | CaO | $SiO_2$ | $Al_2O_3$ | MgO |
|---|---|---|---|---|---|
| E1-R | 37.33 | 28.26 | 9.91 | 18.77 | 5.73 |
| E2-R | 28.36 | 34.35 | 11.55 | 18.91 | 6.83 |
| E3-R | 27.66 | 27.54 | 19.22 | 19.01 | 6.57 |
| E4-R | 25.58 | 35.92 | 10.18 | 18.33 | 9.98 |

**Table 6.** Components of mould flux after TG tests (%).

| composition | $CaF_2$ | $Al_2O_3$ | MgO | $SiO_2$ | CaO | $Na_2O$ |
|---|---|---|---|---|---|---|
| quality | 4.71 | 5.04 | 3.32 | 34.10 | 46.76 | 6.06 |

was also in contrast with the results of the above TG-MS analysis. The XRF tests of remelting electroslag and continuous casting mould flux are shown in Tables 5 and 6.

From the comparison of the results in the above tables with the original composition (Tables 1 and 2), it can be seen that, for the remelted electroslag, the $CaF_2$ and MgO were reduced and the CaO was increased. It was basically consistent with the results of TG-MS analysis and the main volatiles were $CaF_2$ and $MgF_2$. For continuous casting mould flux, in addition to $CaF_2$, the $Na_2O$ also obviously reduced. It was consistent with the TG-MS results, and in addition to the reaction between $CaF_2$ and $Na_2O$, it was found that a small amount of $Na_2O$ was evaporated by calculating the $Na_2O$ weight loss.

The volatilization of fluorine-containing slag in steelmaking is not only difficult to achieve the technical requirements, but also has an impact on the environment considering the toxicity of fluoride. Therefore, it was proposed with the premelted process for fluorine-containing slag [19] that preheat the slag to 1000–1200°C with closed electromagnetic stirring, and then directly participate in the metallurgical process, which can reduce the heat loss, inhibit the volatility of slag and reduce the harm to the environment. The similar effect can also be achieved for the cooled premelted slag after crushing for steelmaking.

## 4. Conclusion

— As typical fluorine-containing slag systems for steelmaking, the remelting electroslag, continuous casting mould flux, and $CaF_2$-CaO-based refining slag can be shown with different volatilization characteristics at high temperature.

— The remelting electroslag volatilized significantly above 1300°C and the volatiles were mainly $CaF_2$, $MgF_2$ with a small amount of $SiF_4$ and $AlF_3$; the continuous casting mould flux volatilization was divided into two stages, in the first stage (500°C~800°C), $CaF_2$ and $Na_2O$ reacted to form NaF, and in the second stage (greater than 1200°C), it was mainly the $CaF_2$ evaporation; for $CaF_2$-CaO-based refining slag, the volatilization was of the most significant at the eutectic point 84% $CaF_2$–16% CaO, and the volatility can be reduced by adding 5% $SiO_2$.

— Take XRF tests of remelting electroslag and continuous casting mould flux after TG tests, and the results were basically consistent with the previous TG-MS analysis. Therefore, this study will be of significance for both slag composition and volatilization characteristics control and metallurgical environmental protection.

Data accessibility. The datasets supporting this article have been uploaded as part of the electronic supplementary material.

Authors' contributions. J.Z. designed the experimental scheme and agreed to be accountable for all aspects of the work in ensuring that questions related to the accuracy or integrity of any part of the work are appropriately investigated and resolved, and will approve of the version to be published; Z.Z. analysed the experimental data and drafted the document; B.Q. and Z.T. made an important contribution to the experiment; Y.C. revised the paper critically for important intellectual content.

Competing interests. We have no competing interests.

Funding. The source of funding for each author: National Natural Science Foundation of China (item no. 51674185) to J.Z. and National Natural Science Foundation of China (item no. 51674186) to Y.C.

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
