## [Reviewer comments · Royal Society Open Science]

Review History

RSOS-200704.R0 (Original submission)

Review form: Reviewer 1

Is the manuscript scientifically sound in its present form?

Yes

Are the interpretations and conclusions justified by the results?

Yes

Is the language acceptable?

Yes

Do you have any ethical concerns with this paper?

No

Have you any concerns about statistical analyses in this paper?

No

Recommendation?

Major revision is needed (please make suggestions in comments)

Comments to the Author(s)

Dear Authors:

Thank you for your consideration to publish your work for Royal Society Open Science journal. The experimental data are very precious. The work investigated the volatilization behavior of fluorine-containing slag systems. However, I recommend major revision because some contradiction can be seen in the descriptions. Please improve the manuscript according to the following comments:

1. Introduction

In the introduction section, the research results of Mills, Chen, Zhao and other scholars are listed. But the difference between this work and the previous scholars' work has not been elaborated. Please describe the difference.

2. Page 4, line 45~47

"According to the previous thermodynamic calculation, the first stage was the reaction of CaF_2 with Na_2O and SiO_2 to generate NaF and SiF_4 gas, and the second stage was mainly the CaF_2 evaporation." The previous research has not been cited, please provide. What is more, please consider whether the experimental conditions of the previous study are the same as this experiment. Otherwise, it should be recalculated and explained in based on this experimental conditions.

3. Page 5, line 7~14

"Different kinds of steel and smelting equipment (VOD and RH) have different requirements for the properties and composition of refining slag." According to the description, authors would like to provide guidance on the VOD and RH slag system though this work. However, during actual steelmaking process, the mass fraction of CaF_2 added in RH or VOD slags is usually below 20%, which is different from the experimental conditions in this article. It is recommended to reconsider the background of this study.

4. Figure 5

The abscissa in the figure is not suitable, it is recommended to revise it to $\text{CaO}/(\text{CaO}+\text{CaF}_2)$.

5. Page 5, line 52~54

The experimental conditions of $\text{CaO}-\text{CaF}_2-\text{SiO}_2$ slag system are not provided, please supplement in the Materials and Methods part.

6. Figure 7

The ordinate in the figure is not right, please revise it.

7. Page6, line 19~22

The author proposed that a small amount of SiO_2 could reduce the volatilization of the $\text{CaO}-\text{CaF}_2-\text{SiO}_2$ slag system, but the experiment results cannot verify this. The CaF_2 contents in R4 and R3 slags are higher than that in R1 slag. The volatilization of R4 and R3 slags are also higher than that of R1 slag. According to these results, it cannot be concluded that the addition of SiO_2 in R1 slag reduce the volatilization of the $\text{CaO}-\text{CaF}_2-\text{SiO}_2$ slags. It is recommended to use TG-MS experiment to investigate the volatiles and volatilization ratio of the $\text{CaO}-\text{CaF}_2-\text{SiO}_2$ slags.

Review form: Reviewer 2

Is the manuscript scientifically sound in its present form?

No

Are the interpretations and conclusions justified by the results?

No

Is the language acceptable?

Yes

Do you have any ethical concerns with this paper?

No

Have you any concerns about statistical analyses in this paper?

No

Recommendation?

Major revision is needed (please make suggestions in comments)

Comments to the Author(s)

The introduction needs to be improved.

The section "3 Materials and Methods" is treated superficially.

A short description of the steel plant.

Insert the name of the metallurgical plant from which the slag samples were taken.

Insert pictures with the three slag samples.

Lines 9-10: Description of the slag samples preparation. Description of the chemical analysis methods.

Specify the names of the "chemical pure reagents".

Lines 23-26: Present the information in the form of a table.

For XRF analysis – specify the type and model of the equipment used and the technical characteristics.

Why the XRF analysis method was chosen?

Specify the steel types and marks from which the slag samples come. Insert a table with the chemical composition of the steels.

Insert the XRF tests for the slag samples before the TG test. Make a comparison of the chemical composition of the slag samples before and after the TG test.

Line 41: Insert the decomposition reaction of Na_2CO_3 .

Lines 45-47 . Insert the chemical reactions.

Lines 52- 53. The paper does not refer to the importance of the study for "metallurgical environmental protection", although this is pointed out in the conclusions. Please insert a scientific discussion and explain.

Specify the concrete measures resulting from the study, regarding the environmental protection in the steelmaking process.

Expand the list of references. E.g: "An Assessment of the Substance Losses from Charge Composition Used to the Steelmaking – Key Factor for Sustainable Steel Manufacturing".

Procedia Manufacturing, vol. 32, 2019, p. 15-21

<https://www.sciencedirect.com/science/article/pii/S2351978919302112>

Decision letter (RSOS-200704.R0)

Dear Professor ZHAO:

Title: The volatilization behavior of typical fluorine-containing slag in steelmaking

Manuscript ID: RSOS-200704

The editor assigned to your manuscript has now received comments from reviewers. We would like you to revise your paper in accordance with the referee and Subject Editor suggestions which can be found below (not including confidential reports to the Editor). Please note this decision does not guarantee eventual acceptance.

Please submit your revised paper before 19-Jun-2020. Please note that the revision deadline will expire at 00.00am on this date. If we do not hear from you within this time then it will be assumed that the paper has been withdrawn. In exceptional circumstances, extensions may be possible if agreed with the Editorial Office in advance. We do not allow multiple rounds of revision so we urge you to make every effort to fully address all of the comments at this stage. If deemed necessary by the Editors, your manuscript will be sent back to one or more of the original reviewers for assessment. If the original reviewers are not available we may invite new reviewers.

RSC Associate Editor:
Comments to the Author:
(There are no comments.)

RSC Subject Editor:
Comments to the Author:
(There are no comments.)

Reviewers' Comments to Author:

Reviewer: 1

Comments to the Author(s)

Dear Authors:

Thank you for your consideration to publish your work for Royal Society Open Science journal. The experimental data are very precious. The work investigated the volatilization behavior of fluorine-containing slag systems. However, I recommend major revision because some contradiction can be seen in the descriptions. Please improve the manuscript according to the following comments:

1. Introduction

In the introduction section, the research results of Mills, Chen, Zhao and other scholars are listed. But the difference between this work and the previous scholars' work has not been elaborated. Please describe the difference.

2. Page 4, line 45~47

"According to the previous thermodynamic calculation, the first stage was the reaction of CaF_2 with Na_2O and SiO_2 to generate NaF and SiF_4 gas, and the second stage was mainly the CaF_2 evaporation." The previous research has not been cited, please provide. What is more, please consider whether the experimental conditions of the previous study are the same as this experiment. Otherwise, it should be recalculated and explained in based on this experimental conditions.

3. Page 5, line 7~14

"Different kinds of steel and smelting equipment (VOD and RH) have different requirements for the properties and composition of refining slag." According to the description, authors would like to provide guidance on the VOD and RH slag system though this work. However, during actual steelmaking process, the mass fraction of CaF_2 added in RH or VOD slags is usually below 20%, which is different from the experimental conditions in this article. It is recommended to reconsider the background of this study.

4. Figure 5

The abscissa in the figure is not suitable, it is recommended to revise it to $\text{CaO}/(\text{CaO}+\text{CaF}_2)$.

5. Page 5, line 52~54

The experimental conditions of $\text{CaO}-\text{CaF}_2-\text{SiO}_2$ slag system are not provided, please supplement in the Materials and Methods part.

6. Figure 7

The ordinate in the figure is not right, please revise it.

7. Page6, line 19~22

The author proposed that a small amount of SiO_2 could reduce the volatilization of the $\text{CaO}-\text{CaF}_2-\text{SiO}_2$ slag system, but the experiment results cannot verify this. The CaF_2 contents in R4 and R3 slags are higher than that in R1 slag. The volatilization of R4 and R3 slags are also higher than that of R1 slag. According to these results, it cannot be concluded that the addition of SiO_2 in R1 slag reduce the volatilization of the $\text{CaO}-\text{CaF}_2-\text{SiO}_2$ slags. It is recommended to use TG-MS experiment to investigate the volatiles and volatilization ratio of the $\text{CaO}-\text{CaF}_2-\text{SiO}_2$ slags.

Reviewer: 2

Comments to the Author(s)

The introduction needs to be improved.

The section "3 Materials and Methods" is treated superficially.

A short description of the steel plant.

Insert the name of the metallurgical plant from which the slag samples were taken.

Insert pictures with the three slag samples.

Lines 9-10: Description of the slag samples preparation. Description of the chemical analysis methods.

Specify the names of the “chemical pure reagents”.

Lines 23-26: Present the information in the form of a table.

For XRF analysis – specify the type and model of the equipment used and the technical characteristics.

Why the XRF analysis method was chosen?

Specify the steel types and marks from which the slag samples come. Insert a table with the chemical composition of the steels.

Insert the XRF tests for the slag samples before the TG test. Make a comparison of the chemical composition of the slag samples before and after the TG test.

Line 41: Insert the decomposition reaction of Na_2CO_3 .

Lines 45-47 . Insert the chemical reactions.

Lines 52- 53. The paper does not refer to the importance of the study for “metallurgical environmental protection”, although this is pointed out in the conclusions. Please insert a scientific discussion and explain.

Specify the concrete measures resulting from the study, regarding the environmental protection in the steelmaking process.

Expand the list of references. E.g: “An Assessment of the Substance Losses from Charge Composition Used to the Steelmaking – Key Factor for Sustainable Steel Manufacturing”.

Procedia Manufacturing, vol. 32, 2019, p. 15-21

<https://www.sciencedirect.com/science/article/pii/S2351978919302112>

Author's Response to Decision Letter for (RSOS-200704.R0)

See Appendix A.

RSOS-200704.R1 (Revision)

Review form: Reviewer 2

Is the manuscript scientifically sound in its present form?

No

Are the interpretations and conclusions justified by the results?

No

Is the language acceptable?

Yes

Do you have any ethical concerns with this paper?

No

Have you any concerns about statistical analyses in this paper?

No

Recommendation?

Major revision is needed (please make suggestions in comments)

Comments to the Author(s)

1. The introduction needs to be improved.
The section "3 Materials and Methods" is treated superficially.
A short description of the steel plant.
2. Insert the name of the metallurgical plant from which the slag samples were taken.
Insert pictures with the three slag samples.
3. Lines 9-10: Description of the slag samples preparation. Description of the chemical analysis methods.
Specify the names of the "chemical pure reagents" (in the section "Materials and methods").
Lines 29-31 (page 5): Present the information in the form of a table.
4. For XRF analysis – specify the type and model of the equipment used and the technical characteristics.
5. Why the XRF analysis method was chosen?
Specify the steel types and marks from which the slag samples come. Insert a table with the chemical composition of the steels.
6. Insert the XRF tests for the slag samples before the TG test.
7. Make a comparison of the chemical composition of the slag samples before and after the TG test.
8. Line 41: Insert the decomposition reaction of Na_2CO_3 .
9. Lines 45-47. Insert the chemical reactions.
Lines 50- 51 (page 8): The paper does not refer to the importance of the study for "metallurgical environmental protection", although this is pointed out in the conclusions. Please insert a scientific discussion and explain.
11. Specify the concrete measures resulting from the study, regarding the environmental protection in the steelmaking process.
10. Expand the list of references.
E.g: "An Assessment of the Substance Losses from Charge Composition Used to the Steelmaking – Key Factor for Sustainable Steel Manufacturing". *Procedia Manufacturing*, vol. 32, 2019, p. 15-21.
The reference is not found in the bibliography.
<https://www.sciencedirect.com/science/article/pii/S2351978919302112>

In conclusion:

The authors treated the observations superficially.

Many observations were ignored. They are marked with red.

Decision letter (RSOS-200704.R1)

Dear Professor ZHAO:

Title: The volatilization behavior of typical fluorine-containing slag in steelmaking
Manuscript ID: RSOS-200704.R1

The editor assigned to your paper has now received comments from reviewers. We would like you to revise your paper in accordance with the referee and Subject Editor suggestions which can be found below (not including confidential reports to the Editor). Please note this decision does not guarantee eventual acceptance.

Please submit a copy of your revised paper before 25-Jul-2020. Please note that the revision deadline will expire at 00.00am on this date. If we do not hear from you within this time then it will be assumed that the paper has been withdrawn. In exceptional circumstances, extensions may be possible if agreed with the Editorial Office in advance. We do not allow multiple rounds of revision so we urge you to make every effort to fully address all of the comments at this stage. If deemed necessary by the Editors, your manuscript will be sent back to one or more of the original reviewers for assessment. If the original reviewers are not available we may invite new reviewers.

RSC Associate Editor:
Comments to the Author:
(There are no comments.)

RSC Subject Editor:
Comments to the Author:
(There are no comments.)

Reviewers' Comments to Author:

Reviewer: 2

Comments to the Author(s)

1. The introduction needs to be improved.

The section "3 Materials and Methods" is treated superficially.

A short description of the steel plant.

2. Insert the name of the metallurgical plant from which the slag samples were taken.

Insert pictures with the three slag samples.

3. Lines 9-10: Description of the slag samples preparation. Description of the chemical analysis methods.

Specify the names of the "chemical pure reagents" (in the section "Materials and methods").

Lines 29-31 (page 5): Present the information in the form of a table.

4. For XRF analysis – specify the type and model of the equipment used and the technical characteristics.

5. Why the XRF analysis method was chosen?

Specify the steel types and marks from which the slag samples come. Insert a table with the chemical composition of the steels.

6. Insert the XRF tests for the slag samples before the TG test.

7. Make a comparison of the chemical composition of the slag samples before and after the TG test.

8. Line 41: Insert the decomposition reaction of Na_2CO_3 .

9. Lines 45-47 . Insert the chemical reactions.

Lines 50- 51 (page 8): The paper does not refer to the importance of the study for "metallurgical environmental protection", although this is pointed out in the conclusions. Please insert a scientific discussion and explain.

11. Specify the concrete measures resulting from the study, regarding the environmental protection in the steelmaking process.

10. Expand the list of references.

E.g: "An Assessment of the Substance Losses from Charge Composition Used to the Steelmaking – Key Factor for Sustainable Steel Manufacturing". *Procedia Manufacturing*, vol. 32, 2019, p. 15-21.

The reference is not found in the bibliography.

<https://www.sciencedirect.com/science/article/pii/S2351978919302112>

In conclusion:

The authors treated the observations superficially.

Many observations were ignored. They are marked with red.

Author's Response to Decision Letter for (RSOS-200704.R1)

See Appendix B.

Decision letter (RSOS-200704.R2)

Dear Professor ZHAO:

Title: The volatilization behavior of typical fluorine-containing slag in steelmaking
Manuscript ID: RSOS-200704.R2

It is a pleasure to accept your manuscript in its current form for publication in Royal Society Open Science. The chemistry content of Royal Society Open Science is published in collaboration with the Royal Society of Chemistry.

RSC Associate Editor
Comments to the Author:
(There are no comments.)

Reviewer(s)' Comments to Author:

Appendix A

Response to Referees Letter

Dear Editor,

Thank you for your letter that the manuscript entitled " **The volatilization behavior of typical fluorine-containing slag in steelmaking**" was received according to the editorial department's opinion.

Thanks to the two reviewers for their comments and suggestions. We have modified this article and answered all the questions one by one, as shown below.

Reviewer #1:

1. In the introduction section, the research results of Mills, Chen, Zhao and other scholars are listed. But the difference between this work and the previous scholars' work has not been elaborated. Please describe the difference.

Response: Thank you for your suggestions. The authors make a summary and evaluation of the previous work and clarify the importance and originality of this work in the text.

2. The previous research has not been cited, please provide. What is more, please consider whether the experimental conditions of the previous study are the same as this experiment.

Response: Thank you for your reminding. The relevant references have been cited and it is clarified that the calculation conditions are the same as the experimental conditions.

3. The mass fraction of CaF₂ added in RH or VOD slags is usually below 20%, which is different from the experimental conditions in this article. It is recommended to reconsider the background of this study.

Response: Thank you for your guidance and suggestions, and it is not appropriate for VOD and RH process considering the mass fraction of CaF₂. The authors have revised the background and statement.

4. The abscissa in the Fig.5 is not suitable, it is recommended to revise it to CaO/(CaO+CaF₂).

Response: Thank you for your guidance and suggestions and the figure has been revised.

5. The experimental conditions of CaO-CaF₂-SiO₂ slag system are not provided, please supplement in the Materials and Methods part.

Response: Thank the Reviewer for the reminder. The experimental conditions have been supplemented and Fig. 7 has been modified.

6. The ordinate in the Fig. 7 is not right, please revise it.

Response: Fig. 7 has been modified in the text.

7. According to these results, it cannot be concluded that the addition of SiO₂ in R1 slag reduce the volatilization of the CaO-CaF₂-SiO₂ slags.

Response: Thank you for your guidance and suggestions and authors add a set of controlled experiments (33%CaF₂-50%CaO-17% SiO₂) to make the discussion more adequate. At the same time, a more detailed discussion of Fig. 7 is carried out in the paper.

Reviewer #2:

1. The introduction needs to be improved.

Response: Thank you for Reviewer's suggestion and the authors make modification and supplement, as shown in the text.

2. Insert the name of the metallurgical plant from which the slag samples were taken.

Response: Thank the Reviewer for the reminder and the steel plant information has been added.

3. Description of the slag samples preparation and the chemical analysis methods.

4. For XRF analysis – specify the type and model of the equipment used and the technical

Response to questions 3 and 4: Thank you for the valuable comments. The XRF analysis method and sample preparation have been supplemented in the text.

5. Why the XRF analysis method was chosen?

Response: X-ray fluorescence (XRF) analysis is one of the most commonly used component analysis methods for slag materials. It has the characteristics of high sensitivity, accurate quantification, fast and convenient. At the same time, the sample preparation method for XRF and the test results are added in the article.

6. Insert the XRF tests for the slag samples before the TG test.

Response: Thank you for your guidance and suggestions. The XRF tests for the slag samples before and after TG tests have been listed in the test.

7. Make a comparison of the chemical composition of the slag samples before and after the TG test.

Response: Thank you for your guidance and suggestions. The comparison of XRF tests for the slag samples before and after TG tests have been added and analysed in the test.

8. Insert the decomposition reaction of Na_2CO_3 .

Response: Thank you for your reminding and the reaction has been added.

9. Insert the chemical reactions.

Response: Thank you for your reminding and the reactions have been added in Table 3.

10. Expand the list of references.

Response: All references have been modified.

11. Specify the concrete measures resulting from the study, regarding the environmental protection in the steelmaking process.

Response: Thanks to the reviewer for this valuable comments. The authors are making relevant analysis on the inhibition of the volatile behavior of fluorine-containing slag and we have supplemented some related research progress in the paper. It has not been discussed too much considering the pre-melting process was not yet mature for all steelmaking slag. We believe that in the near future we will contribute more relevant results to Royal Society Open Science and share with you.

Appendix B

Response to Referees Letter

Dear Editor,

Thank you for your letter that the manuscript entitled " **The volatilization behavior of typical fluorine-containing slag in steelmaking**" was received according to the editorial department's opinion.

Thanks to the two reviewers for their comments and suggestions. We have modified this article and answered all the questions one by one, as shown below.

Reviewer #2:

1. The introduction needs to be improved.

Response: Thank you for Reviewer's suggestion and the authors make modification and supplement, as shown in the text.

2. Insert the name of the metallurgical plant from which the slag samples were taken.

Response: Thank the Reviewer for the reminder and the name and a short description of plant was added.

3. Description of the slag samples preparation and the chemical analysis methods.

4. For XRF analysis – specify the type and model of the equipment used and the technical

Response to questions 3 and 4: Thank you for the valuable comments. The XRF analysis method and sample preparation have been supplemented in the text.

5. Insert pictures with the three slag samples.

Response: the remelting electroslag and continuous casting mold flux pictures were supplemented, and the traditional refining slag was prepared under the different composition ratio.

6. Why the XRF analysis method was chosen?

Response: X-ray fluorescence (XRF) analysis is one of the most commonly used component analysis methods for slag materials. It has the characteristics of high sensitivity, accurate quantification, fast and convenient. At the same time, the sample preparation method for XRF and the test results are added in the article.

7. Insert the XRF tests for the slag samples before the TG test.

Response: Thank you for your guidance and suggestions. The XRF tests for the slag samples before and after TG tests have been listed in the test.

8. Present the reagent information in the form of a table.

Response: the reagent information was added in Table 1.

9. Make a comparison of the chemical composition of the slag samples before and after the TG test.

Response: Thank you for your guidance and suggestions. The comparison of XRF tests for the slag samples before and after TG tests have been added and analysed in the test.

10. Insert the decomposition reaction of Na₂CO₃.

Response: Thank you for your reminding and the reaction has been added.

11. Insert the chemical reactions.

Response: Thank you for your reminding and the reactions have been added in Table 3.

12. Expand the list of references.

Response: All references have been modified.

13. Specify the concrete measures resulting from the study, regarding the environmental protection in the steelmaking process.

Response: Thanks to the reviewer for this valuable comments. The authors are making relevant analysis on the inhibition of the volatile behavior of fluorine-containing slag and we have supplemented some related research progress in the paper. It has not been discussed too much considering the pre-melting process was not yet mature for all steelmaking slag. We believe that in the near future we will contribute more relevant results to Royal Society Open Science and share with you.